# CREATIDESIGN: A UNIFIED MULTI-CONDITIONAL DIFFUSION TRANSFORMER FOR CREATIVE GRAPHIC DESIGN

**Hui Zhang**[1,2,3,*]   **Dexiang Hong**[3,4,*]   **Maoke Yang**[3]   **Yutao Cheng**[3]   **Zhao Zhang**[3]
**Weidong Chen**[4]   **Jie Shao**[3]   **Xinglong Wu**[3]   **Zuxuan Wu**[1,2,†]   **Yu-Gang Jiang**[1,2,†]
[1]Institute of Trustworthy Embodied AI, Fudan University
[2]Shanghai Key Laboratory of Multimodal Embodied AI
[3]Bytedance Intelligent Creation
[4]School of Information Science and Technology, University of Science and Technology of China

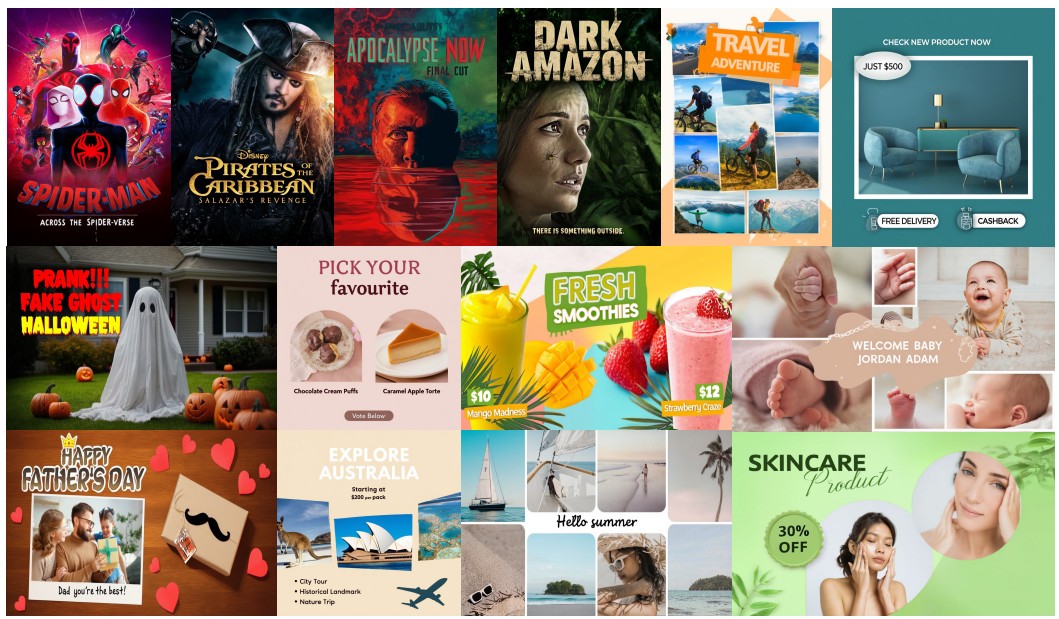

Figure 1: CreatiDesign generates high-quality graphic designs based on user-provided image assets and semantic layouts, covering a wide range of categories such as movie posters, brand promotions, product advertisements, and social media content.

## ABSTRACT

Graphic design plays a vital role in visual communication across advertising, marketing, and multimedia entertainment. Prior work has explored automated graphic design generation using diffusion models, aiming to streamline creative workflows and democratize design capabilities. However, complex graphic design scenarios require accurately adhering to design intent specified by multiple heterogeneous user-provided elements (*e.g.* images, layouts, and texts), which pose multi-condition control challenges for existing methods. Specifically, previous single-condition control models demonstrate effectiveness only within their specialized domains but fail to generalize to other conditions, while existing multi-condition methods often lack fine-grained control over each sub-condition and compromise overall compositional harmony. To address these limitations, we introduce CreatiDesign, a systematic solution for automated graphic design covering both model architecture and dataset construction. First, we design a unified multi-condition driven architecture that enables flexible and precise integration

* Equal contribution. † Corresponding author.

of heterogeneous design elements with minimal architectural modifications to the base diffusion model. Furthermore, to ensure that each condition precisely controls its designated image region and to avoid interference between conditions, we propose a multimodal attention mask mechanism. Additionally, we develop a fully automated pipeline for constructing graphic design datasets, and introduce a new dataset with 400K samples featuring multi-condition annotations, along with a comprehensive benchmark. Experimental results show that CreatiDesign outperforms existing models by a clear margin in faithfully adhering to user intent.

# 1 INTRODUCTION

Graphic design (Jobling & Crowley, 1996) is a fundamental vehicle for visual communication, affective perception, and brand identity across advertising, marketing, and multimedia entertainment.

Recently, diffusion models (Ho et al., 2020; Dhariwal & Nichol, 2021) have achieved remarkable advances, especially in text-to-image generation (stability.ai, 2024; Labs, 2024), which can produce visually compelling and semantically rich images. Leveraging these models to automate graphic design—thereby streamlining creative workflows and democratizing design capabilities—has attracted increasing attention (Gao et al., 2025a; Chen et al., 2025; Peng et al., 2025).

However, as illustrated in Figure 2, graphic design generation poses unique challenges because it requires the precise control and harmonious arrangement of multiple heterogeneous elements, typically comprising three categories: I) **Primary visual elements**, which act as visual focal points and convey the central theme (*e.g.* product subjects, provided in image format); II) **Secondary visual elements**, which offer contextual support and enrich the composition (*e.g.* decorative objects, specified by semantic description and position in a layout); and III) **Textual elements**, which directly convey essential information (*e.g.* slogans or product names, also provided as layout). This multi-element nature introduces multi-condition control requirements for diffusion models, as it demands both semantic and spatial fidelity to users' design intent.

While several works have explored unleashing the potential of diffusion models for automatic graphic design generation, three major challenges remain unresolved: I) **How to integrate multiple heterogeneous conditions in a unified manner.** Previous expert models are typically tailored for only a single type of condition, and often fail to follow other conditions. As illustrated in Figure 2, image-driven models Wang et al. (2024b); Wu et al. (2025); Labs (2025) focus exclusively on aligning with primary visual elements, whereas layout-driven models Peng et al. (2025); Ma et al. (2025b); Zhang et al. (2024) are limited to following the semantic descriptions and spatial arrangements of secondary visual or textual elements. Such biased capability often leads to reduced fidelity to user intent, as highlighted by the red and purple masks. II) **How to preserve fine-grained controllability for each condition while achieving harmonious compositions.** Existing multi-condition approaches (Xiao et al., 2024; Gao et al., 2025a; goo, 2025; ope, 2025) lack accurate control over each sub-condition and fail to effectively coordinate all elements, resulting in outputs that do not faithfully reflect user design intent. III) **How to construct large-scale, multi-element graphic design datasets in an automated manner.** Ready-to-use graphic design datasets with fine-grained, multi-condition annotations remain scarce, which naturally prevents models from learning design capabilities.

To this end, we propose CreatiDesign, a systematic solution for intelligent graphic design generation that addresses the aforementioned challenges through the following components: I) **Unified multi-condition driven architecture.** CreatiDesign preserves the strong generative capabilities of text-to-image diffusion models while unlocking their potential for graphic design with minimal architectural modifications. Specifically, the native image encoder embeds the multi-subject image condition into the latent space, while the semantic layout is processed by extracting textual features with the text encoder and fusing them with positional information. After encoding all modalities into a unified feature space, native multimodal attention (MM-Attention) is applied to enable deep integration and interaction across modalities. This allows for unified and flexible multi-condition control over the generated content. II) **Efficient Multi-Condition Coordination.** To ensure that each heterogeneous condition precisely controls its designated image regions and to avoid mutual interference that could compromise the unique characteristics of each condition, we introduce carefully designed attention masks to regulate the interaction scope of each modality within the multimodal attention mechanism. This design enables each condition to independently and efficiently control its target region, while

maintaining high overall compositional harmony. III) **Automated Dataset Construction Pipeline.** We develop a fully automated pipeline for constructing graphic design datasets. This pipeline consists of design theme generation and rendering, conditional image generation, and multi-element annotation and filtering. As a result, we construct a training dataset containing 400K design samples with multi-condition annotations, along with a comprehensive benchmark for rigorous evaluation.

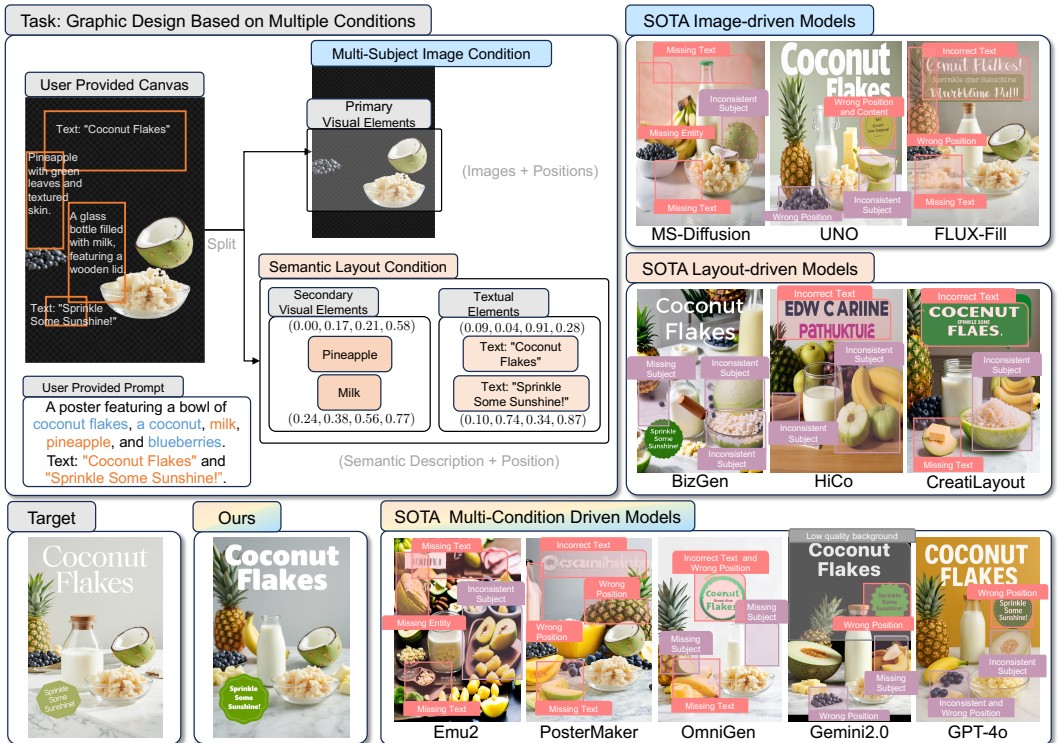

Figure 2: **An overview of our motivation**. Graphic design is a multi-condition driven generation task that requires the precise and harmonious arrangement of heterogeneous elements, including primary visual elements (provided as images with positions), as well as secondary visual and textual elements (both specified by semantic descriptions and positions). Previous methods either support only a single type of condition (*e.g.* image-driven or layout-driven models) or lack accurate control over each sub-condition(*e.g.* multi-condition driven models), resulting in failure to strictly adhere to user design intent, as highlighted by the red and purple masks.

## 2 RELATED WORK

### 2.1 TEXT-TO-IMAGE GENERATION

Text-to-image (T2I) generation (Rombach et al., 2022; Podell et al., 2024; Saharia et al., 2022; Chen et al., 2024c; Li et al., 2024) aims to generate visual content from textual descriptions, and has achieved remarkable progress in both visual quality and semantic alignment. Recent advances, such as SD3 series (Esser et al., 2024; stability.ai, 2024), CogView4 (THU, 2025), FLUX.1 (Labs, 2024), HiDream (HiD, 2025), and Seedream series (Gong et al., 2025; Gao et al., 2025b), have pushed the frontier further by leveraging Multimodal Diffusion Transformer architectures (MM-DiT). Despite these advances, existing T2I models still struggle with fine-grained controllability, particularly in scenarios where users wish to specify precise subject identities or detailed compositional layouts.

### 2.2 CONTROLLABLE IMAGE GENERATION

To achieve precise control, a variety of conditional image generation paradigms have been proposed, including subject-driven (Ruiz et al., 2023; Cai et al., 2024; Tan et al., 2024; Shin et al., 2024; Zhu et al., 2025; Labs, 2025; Wu et al., 2025; Wang et al., 2024b), layout-driven (Li et al., 2023; Wang et al., 2024c; Zhou et al., 2024; Feng et al., 2024; Zhang et al., 2024; Peng et al., 2025;

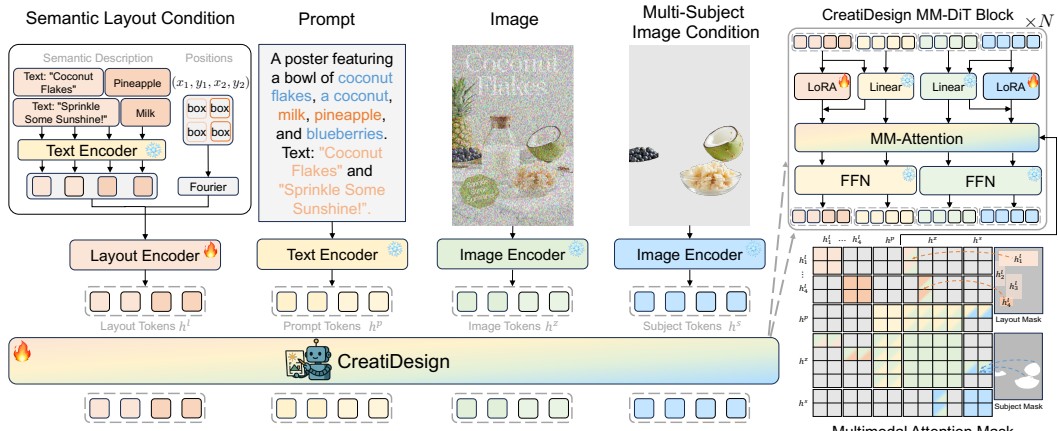

Figure 3: **An overview of the architecture.** CreatiDesign integrates subjects and semantic layout conditions through native multimodal attention. Multimodal attention mask ensures that each condition precisely controls its designated image regions while preventing leakage between conditions.

Zhou et al., 2025; Ma et al., 2025b), and so on. These expert models excel at controlling specific conditions—such as preserving the visual characteristics of the provided subjects or adhering to layout specifications—but often fail to follow other conditions. In response, multi-condition driven frameworks (Sun et al., 2024; Xiao et al., 2024; goo, 2025; ope, 2025; Wang et al., 2025a; Qin et al., 2023; Hu et al., 2023; Zhao et al., 2023; Ran et al., 2024; Yang et al., 2024) have been introduced to jointly handle heterogeneous user-provided conditions. However, these unified approaches often lack accurate control over each sub-condition.

## 2.3 AUTOMATIC GRAPHIC DESIGN

Several works (Gao et al., 2025a; Wang et al., 2025b; Chen et al., 2025; Pu et al., 2025; Ma et al., 2025a; Wang et al., 2024a; Liu et al., 2024; Chen et al., 2024b; Tuo et al., 2024) have attempted to automate graphic design generation, aiming to streamline creative workflows and democratize design capabilities. However, automatic graphic design introduces distinct challenges beyond general text-to-image or controllable image generation, requiring models to precisely preserve user-specified subjects, align secondary visual and textual elements with detailed semantic and spatial constraints, and maintain overall visual coherence. Despite recent progress, most existing methods struggle to meet all these demands simultaneously. This underscores the need for a unified, highly controllable, and harmonious solution, which is exactly the goal of this paper.

## 3 METHOD

### 3.1 PROBLEM FORMULATION

This paper focuses on the task of graphic design generation, where each design typically comprises multiple heterogeneous elements provided by the user, such as primary visual elements, secondary visual elements, and textual elements, as illustrated in Figure 2. The key challenge is to accurately and harmoniously integrate these user-specified elements—each representing distinct aspects of user intent—into the generated image. Formally, the task can be defined as: $I_g = f(P, I_s, L)$, where $I_g$ denotes the generated image, $P$ is the global prompt describing the overall image, $I_s$ represents the multi-subject image condition (*i.e.*, a set of primary visual elements). $L$ denotes the semantic layout condition, which consists of $n$ elements, partitioned into two categories: secondary visual elements and textual elements. Each layout element is defined by a pair $(d_i, b_i)$, where $d_i$ is the semantic description and $b_i$ is the spatial position (bounding box), formally expressed as:

$$L = \{l_i = (d_i, b_i)\}_{i=0}^n, \quad l_i \in \{\text{secondary visual element, textual element}\}. \tag{1}$$

In the following sections, we will introduce the key parts of CreatiDesign in detail.

## 3.2 Unified Multi-Condition Driven Architecture

In MM-DiT-based text-to-image models (*e.g.* FLUX.1 (Labs, 2024)), a text encoder (*e.g.* T5 (Raffel et al., 2020)) is employed to tokenize and encode the input prompt into a sequence of text tokens, denoted as $h_p$. Concurrently, an image encoder (*e.g.* VAE (Kingma, 2013)) is utilized to encode the ground-truth image into a latent representation $z$, which is subsequently partitioned into patches to obtain image tokens, denoted as $h_z$. These text and image tokens are then fed into MM-Attention, which facilitates rich interactions between the textual and visual modalities, thereby enabling precise control over the image content. Our approach aims to retain the strong capabilities of T2I models while unlocking their potential for graphic design with minimal architectural modifications, as illustrated in Figure 3.

**Tokenize Multi-Subject Image Condition.** We first pad the multi-subject image condition with a background color (*e.g.* gray) and encode it using the native VAE. The encoded latent representation is then partitioned into patches to obtain the subject tokens $h_s$.

**Tokenize Semantic Layout Condition.** For each element $l_i = (d_i, b_i)$ in the semantic layout condition, we utilize the native T5 text encoder to extract the semantic feature $h_i^d$ from $d_i$. For the bounding box $b_i$, we apply Fourier positional encoding (Mildenhall et al., 2021; Li et al., 2023) to obtain the spatial feature $h_i^b$. The final layout token $h_i^l$ is obtained by concatenating $h_i^d$ and $h_i^b$ along the feature dimension, followed by a layout encoder (*i.e.* MLP): $h_i^l = \text{MLP}(\text{Concat}(h_i^d, h_i^b))$. In this way, layout tokens integrate semantic and spatial information.

**Integrate Multi-Condition.** After encoding the prompt, noise image, multi-subject image condition, and semantic layout condition into tokens, denoted as $h^p, h^z, h^s, h^l$, we concatenate them along the token dimension and feed the token sequence into a stack of MM-DiT Blocks. Each Block consists of linear projection layers (for Q, K, V), multimodal attention (MM-Attention), and feed-forward networks (FFN). Each type of tokens is linearly projected into its corresponding query, key, and value spaces: $Q^*, K^*, V^* = \text{Linear}(h^*)$, where $*$ denotes the modality (layout, prompt, image, or subject). For the layout tokens $h^l$ and subject tokens $h^s$, we further adapt their representations using LoRA modules deployed on the linear layer and adaptive layer normalization (AdaLN), enabling efficient fine-tuning and alignment. The multimodal attention is then computed as:

$$h^{l\prime}, h^{p\prime}, h^{z\prime}, h^{s\prime} = \text{Attention}([\mathbf{Q}^l, \mathbf{Q}^p, \mathbf{Q}^z, \mathbf{Q}^s], [\mathbf{K}^l, \mathbf{K}^p, \mathbf{K}^z, \mathbf{K}^s], [\mathbf{V}^l, \mathbf{V}^p, \mathbf{V}^z, \mathbf{V}^s]). \quad (2)$$

This design enables multiple conditions to control the image content. To avoid positional embedding conflicts, such as between the noise image and image condition, or between the prompt and layout condition, we adopt positional encoding shifts to the image and layout condition tokens (Tan et al., 2024) to ensure clear separation in the token space. Overall, this architecture empowers the text-to-image model with multi-condition control capabilities through minimal architectural modifications.

## 3.3 Collaborative Multi-Condition Control

Multi-condition driven methods may suffer from degraded controllability over each sub-condition. We attribute this to the fact that the sub-condition is not precisely bound to its corresponding image region and that there is semantic leakage among sub-conditions. To address this, we introduce a multimodal attention mask within our architecture, consisting of a layout mask and a subject mask.

**Layout Attention Mask.** Given the user-specified bounding box $b_i$ for each semantic description $d_i$, we can precisely locate the target image region. Inspired by (Chen et al., 2024a), we construct a layout mask such that each layout token $h_i^l$ is only allowed to attend to and be attended by the image tokens $h_i^z$ within its corresponding bounding box. This explicit attention modulation enhances the spatial controllability. Furthermore, we block interactions among layout tokens themselves, between layout tokens and subject tokens, and between layout tokens and prompt tokens, to prevent semantic leakage and to ensure that each layout token retains its unique characteristics.

**Subject Attention Mask.** Based on the user-provided multi-subject image, we extract the spatial location of each subject to form a subject mask. Each subject token $h_i^s$ is only permitted to interact bidirectionally with the image tokens $h_i^z$ within its own mask region, thereby achieving precise subject injection. In addition, to preserve the integrity and distinctive features of the subject token $h^s$, we block its interactions with all irrelevant tokens, including layout tokens $h^l$, prompt tokens $h^p$, and image tokens outside the target region of $h^s$.

With the proposed multimodal attention masks, CreatiDesign allows each condition to precisely and independently control its targeted image region without semantic leakage, thereby producing controllable and harmonious graphic designs that closely match user intent.

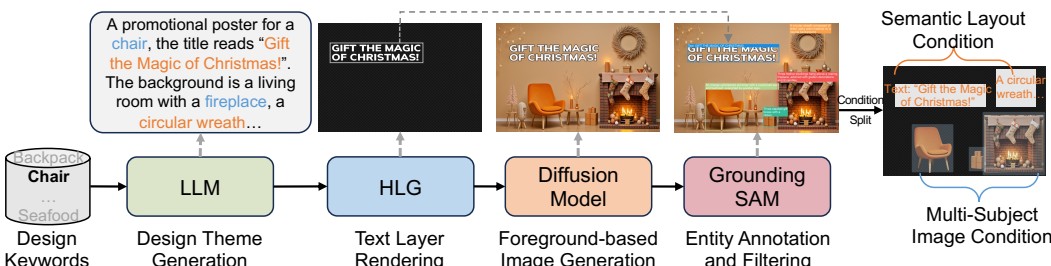

Figure 4: Automated pipeline for graphic design dataset construction.

# 4 GRAPHIC DESIGN DATASETS AND BENCHMARK

## 4.1 GRAPHIC DESIGN DATASETS

We propose a fully automatic dataset construction pipeline, as shown in Figure 4, to address the scarcity of graphic design datasets with fine-grained, multi-condition annotations.

**Design Theme Generation.** Based on a design keywords bank covering common graphic design elements (*e.g.* furniture, food, clothing *etc.*), we prompt a large language model (LLM, *e.g.* GPT-4) to act as a professional designer and generate design themes that include descriptions of primary visual elements, secondary visual elements, and textual elements.

**Text Layer Rendering.** Based on the design theme, we follow the Hierarchical Layout Generation (Cheng et al., 2025) (HLG) paradigm to generate a layout protocol of textual elements and a detailed background description. A rendering engine then converts the layout protocol into an RGBA image with accurately positioned foreground text.

**Foreground-based Image Generation.** To generate a visually coherent graphic design image, we draw inspiration from LayerDiffuse (Zhang & Agrawala, 2024) and develop a foreground-conditioned image generation model. Here, the RGBA text layer serves as the foreground, while the background is generated based on the aforementioned description. Specifically, we incorporate foreground-LoRA and background-LoRA modules into FLUX.1-dev and employ attention sharing to ensure seamless integration of foreground and background elements.

**Entity Annotation.** We use GroundingSAM2 (ide, 2024) to obtain bounding boxes and segmentation masks for all entities in the generated image. A vision-language model (OpenBMB, 2024) (VLM) is then employed to generate fine-grained descriptions for each entity. Entities are categorized as either primary or secondary visual elements. All primary visual elements are aggregated to form the multi-subject image condition, while secondary visual elements, together with the textual elements from the layout protocol, constitute the semantic layout condition.

Based on this automatic data construction pipeline, we synthesize graphic design samples at scale, alleviating the data bottleneck for model training. As a result, we construct a new dataset of 400K samples with annotations for various conditions.

## 4.2 GRAPHIC DESIGN BENCHMARK

To comprehensively evaluate graphic design generation under multiple conditions, we further construct a rigorous benchmark consisting of 1,000 carefully curated samples. This benchmark is designed to assess whether the generated results faithfully align with user intent—a critical requirement in practical graphic design scenarios.

The evaluation focuses on two key aspects, each with dedicated metrics: I) **Multi-Subject Preservation.** When given multi-subject image conditions (*i.e.* primary visual elements), it is crucial to

strictly preserve the unique characteristics of each subject in the generated image. To quantify this, we measure the similarity between each subject and its corresponding region (obtained via bounding box priors or detected by GroundingDINO (Liu et al., 2023)) in the generated image using both CLIP (Radford et al., 2021) similarity (CLIP-I) and DINO (Oquab et al., 2023) similarity (DINO-I) scores. We further aggregate the DINO scores of all subjects by multiplication, denoted as M-DINO (Wang et al., 2024b). Unlike averaging, M-DINO is more sensitive to the failure of any single subject, providing a stricter assessment of subject preservation. II) **Semantic Layout Alignment.** For the semantic layout condition, specifying the positions and attributes of secondary visual elements and textual content, we assess alignment at spatial and semantic levels. For secondary visual elements, we employ a vision-language model in a Visual Question Answering manner to assess the spatial, color, textual, and shape attributes of each entity in the generated image (Zhang et al., 2024; Wu et al., 2024). For textual elements, we use PaddleOCR (pad, 2025) to detect text and calculate sentence accuracy (Sen. Acc), normalized edit distance (NED; *i.e.* 1 minus the edit distance) (Gao et al., 2025a), and IoU (spatial score) between detected and ground-truth texts.

## 5 EXPERIMENTS

### 5.1 EXPERIMENTAL SETUP

**Dataset.** We train our models on the 400K synthetic graphic design samples described in Section 4.1. The validation set contains 1,000 samples, covering diverse numbers of primary visual subjects and semantic layout annotations, enabling thorough evaluation of multi-condition controllability.

**Evaluation Metrics.** As described in Section 4.2, we evaluate model performance from two perspectives—multi-subject preservation and semantic layout alignment—to assess whether the generated designs accurately fulfill user intent. Additionally, to evaluate overall image quality, we report IR Score (Xu et al., 2023) and PickScore (Kirstain et al., 2023), which jointly capture prompt adherence, visual appeal, and compositional harmony across the entire image.

**Implementation Details.** We fine-tune FLUX.1-dev using LoRA with 256 rank, introducing 491.5M extra parameters (4.1% of FLUX's 12B). We employ the AdamW optimizer with a fixed learning rate of 1e-4, training for 100,000 steps with a batch size of 8 on 8 H20-96G GPUs over 4 days. We adopt a resolution bucketing strategy during training to support variable image sizes. The image condition is set to half the target image size; each layout description is capped at 30 tokens, with up to 10 layouts per image.

### 5.2 COMPARISON WITH PRIOR WORKS

**Baseline Methods.** We compare CreatiDesign with three types of previous SOTA models: multi-subject image-driven models (Wu et al., 2025; Wang et al., 2024b; Labs, 2025), semantic layout-driven models (Zhang et al., 2024; Ma et al., 2025b; Peng et al., 2025; Tuo et al., 2024), and multi-condition driven models (Sun et al., 2024; goo, 2025; Xiao et al., 2024; Gao et al., 2025a).

**Quantitative Comparison.** As shown in Table 1, specialist models excel primarily on their targeted control conditions—image-driven models can preserve subjects, while layout-driven models can follow semantic layout control—but perform poorly when handling other conditions. Conversely, previous multi-condition models often lack fine-grained control over each sub-condition, resulting in lower subject preservation and semantic alignment. In contrast, CreatiDesign achieves precise, balanced control across all conditions, as reflected in its top-tier performance across every sub-condition and clear lead in average scores. Remarkably, this advanced graphic design capability is achieved with minor architectural modifications to the base model FLUX.1-dev and only 4.1% extra parameters were introduced, demonstrating both effectiveness and efficiency.

**Qualitative Comparison.** To further illustrate the advantages of CreatiDesign, Figure 7 presents qualitative comparisons on challenging cases with multiple subjects and complex layouts. Existing SOTA methods—including multi-condition driven models and single-condition experts—consistently fall short in faithfully fulfilling user intent. Previous multi-condition models exhibit limited precision in controlling sub-conditions, resulting in misplaced or inconsistent subjects (highlighted by purple masks), as well as content or spatial misalignment in the layout (highlighted by red masks). Layout-driven models like BizGen (Peng et al., 2025) can follow the layout but struggle with subject

Table 1: **Quantitative Results**. We compare CreatiDesign with three types of previous SOTA models: multi-subject image-driven models, semantic layout-driven models, and multi-condition driven models. The best results are shown in **bold**, and the top-3 results are highlighted. Our proposed method significantly enhances the graphic design capabilities of the baseline, achieves top-tier performance across all metrics, and shows a clear lead in average score.

| | Multi-Subject Preservation | | | Semantic Layout Alignment | | | | | | | Image Quality | | Avg. |
| | Primary Visual Elements | | | Secondary Visual Elements | | | | Textual Elements | | | | | |
| | CLIP-I | DINO-I | M-DINO | Spatial | Color | Textual | Shape | Spatial | Sen. Acc | NED | IR | PickScore | |
|---|---|---|---|---|---|---|---|---|---|---|---|---|---|
| UNO | 77.97 | 47.88 | 20.79 | 53.10 | 43.44 | 42.62 | 41.30 | 11.47 | 40.87 | 74.51 | **61.06** | **21.67** | 44.72 |
| MS-Diffusion | 84.75 | 74.13 | 44.34 | 49.54 | 33.37 | 34.89 | 34.79 | 1.01 | 0.00 | 10.21 | 46.64 | 21.03 | 36.23 |
| FLUX.1-Fill | 90.79 | 87.32 | 69.05 | 67.55 | 57.48 | 56.26 | 55.75 | 12.48 | 12.07 | 56.69 | 40.74 | 20.71 | 52.24 |
| CreatiLayout | 78.41 | 55.54 | 25.31 | 77.42 | 63.07 | 62.67 | 60.21 | 18.59 | 12.27 | 74.21 | 59.92 | 21.04 | 50.72 |
| HiCo | 72.45 | 34.47 | 11.59 | 79.69 | 61.48 | 61.17 | 60.17 | 1.01 | 0.00 | 14.98 | -36.16 | 19.58 | 31.70 |
| BizGen | 79.86 | 53.08 | 22.93 | 79.84 | 62.96 | 62.86 | 61.01 | 50.44 | 75.89 | 94.61 | 37.43 | 21.48 | 58.53 |
| AnyText2 | 74.68 | 34.86 | 12.22 | 36.63 | 27.58 | 27.16 | 26.53 | 53.95 | 9.56 | 48.26 | -25.95 | 20.24 | 28.81 |
| Emu2 | 73.96 | 45.17 | 19.92 | 60.81 | 45.37 | 46.20 | 44.06 | 0.20 | 0.00 | 13.81 | -1.84 | 20.18 | 30.65 |
| PosterMaker | 90.45 | **87.72** | **69.56** | 56.36 | 45.37 | 44.25 | 41.61 | 28.42 | 0.70 | 28.62 | 31.62 | 20.31 | 45.42 |
| OmniGen | 82.15 | 58.83 | 30.86 | 53.92 | 44.35 | 44.46 | 41.40 | 8.24 | 6.72 | 49.12 | 22.94 | 20.62 | 38.63 |
| Gemini2.0 | 81.46 | 57.23 | 29.68 | 59.41 | 52.29 | 52.49 | 50.36 | 16.60 | 71.38 | 88.71 | 28.52 | 21.23 | 50.78 |
| FLUX.1-dev | 75.93 | 44.59 | 17.76 | 60.02 | 47.1 | 46.19 | 44.76 | 13.25 | 57.95 | 81.52 | 59.45 | 21.48 | 47.50 |
| **CreatiDesign** | 89.39 | 86.48 | 65.75 | 78.94 | **66.02** | **66.94** | **65.82** | 56.90 | **78.30** | 94.68 | 60.02 | 21.49 | **69.28** |
| *vs. Baseline* | **+13.46** | **+41.89** | **+47.99** | **+18.92** | **+18.92** | **+20.75** | **+21.06** | **+43.65** | **+20.35** | **+13.16** | **+1.17** | **+0.01** | **+21.78** |

consistency. Image-driven models such as FLUX.1-Fill (Labs, 2025) can preserve primary elements but often misplace or incorrectly render textual elements. In contrast, CreatiDesign consistently preserves the identity and position of all primary subjects, precisely aligns secondary and textual elements within the layout, and ensures overall compositional harmony.

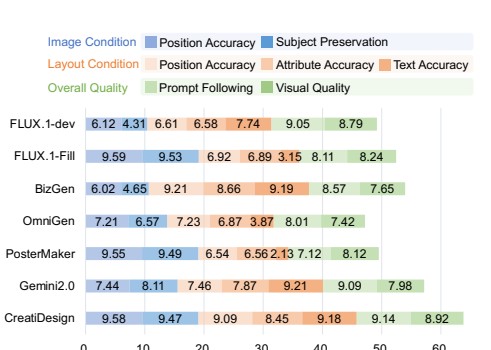

Figure 5: **User Study.** CreatiDesign demonstrates top-tier performance in preserving multi-subject characteristics, strictly following semantic layout conditions, and achieving high overall image quality.

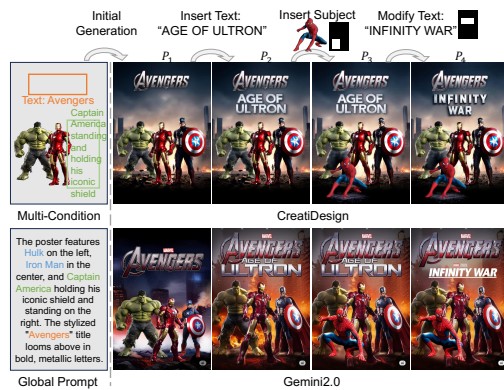

Figure 6: **Comparison on Loop Editing.** CreatiDesign precisely follows editing commands and maintains high consistency in non-edited areas. In contrast, Gemini2.0 frequently introduces unwanted attribute changes to subjects or text.

**User Study.** To comprehensively assess the practical effectiveness of CreatiDesign, we conducted a user study involving feedback from both professional designers and general users. Specifically, we solicited 50 evaluation reports on 30 diverse graphic design samples, comparing our method with several state-of-the-art baselines. As illustrated in Figure 5, participants rated the generated designs on a scale of 1 to 10 across multiple criteria, including adherence to multi-subject image conditions (position accuracy and subject preservation), alignment with semantic layout conditions (position accuracy, attribute accuracy and text accuracy), and overall perceptual quality (prompt following and visual quality). The statistical results demonstrate that CreatiDesign outperforms

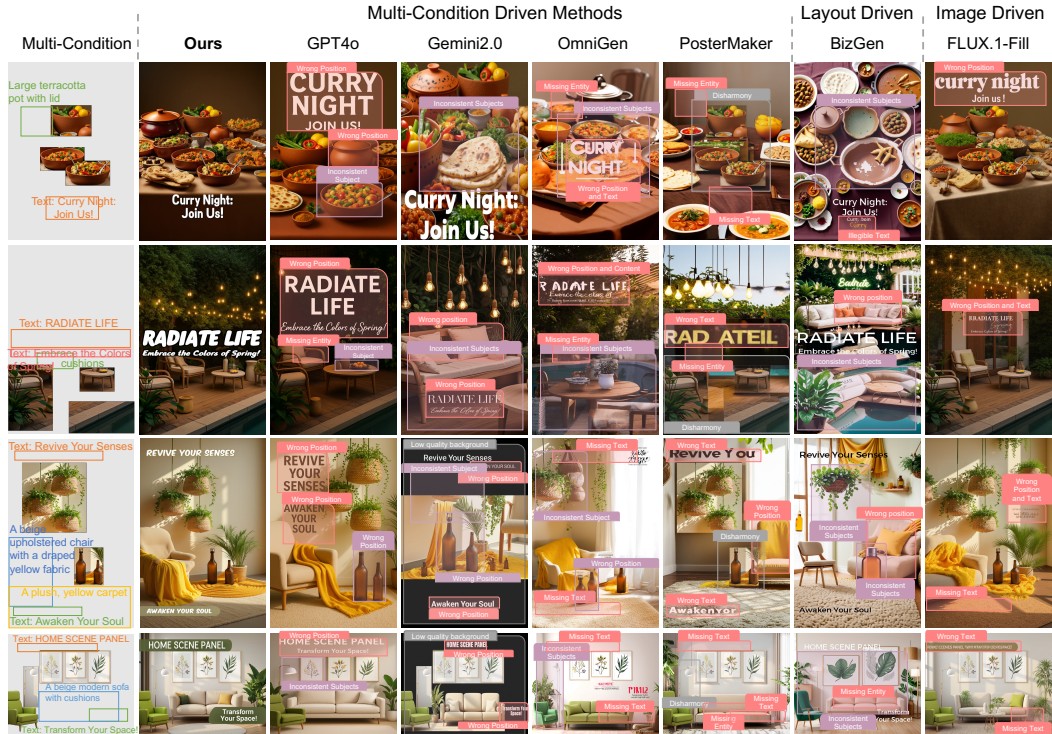

Figure 7: **Quantitative Results.** Compared with previous multi-condition or single-condition models, CreatiDesign demonstrates stricter adherence to user intent, including high subject preservation and precise layout alignment. Purple masks: inconsistent or mispositioned subjects. Red mask: entities with incorrect semantics or locations. Gray mask: disharmonious background or foreground regions.

previous methods in fine-grained controllability and overall visual appeal, delivering superior user satisfaction in real-world graphic design scenarios.

## 5.3 FREE LUNCH: EXPANDING TO EDITING TASKS

As illustrated in Figure 6, CreatiDesign naturally extends beyond graphic design to a wide range of editing tasks without extra retraining. We demonstrate this capability via editing a series of movie posters. Initially, the user provides a global prompt, a multi-subject image condition (*e.g.* Hulk and Iron Man), and a semantic layout specifying elements and their spatial positions (*e.g.* Captain America and the "Avengers" title). CreatiDesign generates a high-quality poster $P_1$ that precisely adheres to these controls. Subsequently, a sequence of editing operations is performed: first, by treating the previously generated poster $P_1$ as the new image condition and introducing a new text element "AGE OF ULTRON" with its desired position, CreatiDesign seamlessly inserts this subtitle to produce $P_2$; Next, by combining the Spider-Man image and its insertion mask with $P_2$ as the image condition, CreatiDesign generates $P_3$, achieving seamless integration of the new subject while preserving character fidelity and overall visual harmony; finally, by combining $P_3$ with the mask of the edited region as the image condition, the subtitle is modified to "INFINITY WAR" ($P_4$). Throughout these editing processes, CreatiDesign consistently maintains subject identity, achieves accuracy layout control and overall visual harmony. In contrast, strong baselines such as Gemini2.0 frequently fail to preserve non-edited regions during sequential edits, often resulting in unwanted attribute changes to subjects or text, highlighting a lack of strict adherence to user intent.

## 5.4 ABLATION STUDY

Table 2 and Figure 8 evaluate the contributions of the three key components—Layout Encoder (LE), Layout Attention Mask (LAM), and Subject Attention Mask (SAM)—to the performance of CreativeDesign from quantitative and qualitative perspectives, respectively. The LE fuses the semantic features of the textual description with Fourier-encoded positional features and further aligns them into layout tokens; removing LE leads to a clear drop in the accuracy of generated text. The

layout attention mask enables fine-grained spatial control by explicitly restricting each layout element to modulate only its designated image region and preventing semantic leakage across layout elements; removing LAM leads to imprecise placement of elements and increased confusion across different layout regions, as demonstrated by the decrease in spatial alignment and attribute accuracy. Similarly, the subject attention mask ensures that each subject token only interacts with its corresponding image region and blocks interference from global prompts and layout conditions. Without SAM, we observe the degradation in subject consistency, such as the incorrect digits on clocks or altered popcorn color. These results validate the effectiveness of each component in achieving faithful and controllable graphic design generation.

Table 2: Ablation study: quantitative analysis of key components in CreatiDesign.

| | Subject Preservation | | Semantic Layout Alignment | | | |
| | Visual Elements | | Visual Elements | | Textual Elements | |
| | DINO | M-DINO | Spatial | Attribute | Spatial | Sen. Acc |
|---|---|---|---|---|---|---|
| **CreatiDesign** | **86.48** | **65.75** | 78.94 | 66.26 | **56.90** | **78.30** |
| w/o LE | 85.10 | 62.96 | **80.99** | **69.24** | 52.42 | 12.13 |
| w/o LAM | 85.79 | 64.28 | 66.94 | 56.19 | 20.16 | 68.41 |
| w/o SAM | 85.70 | 64.14 | 75.99 | 64.90 | 56.92 | 76.84 |

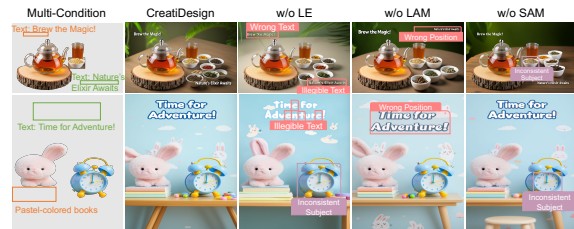

Figure 8: Qualitative Results of Ablation Study.

## 6 CONCLUSION

In this paper, we presented CreatiDesign, a systematic solution that empowers diffusion transformers for intelligent and highly controllable graphic design generation. We designed a unified multi-condition driven architecture that seamlessly integrates heterogeneous design elements. Furthermore, we proposed a multimodal attention mask mechanism to ensure that each condition precisely controls its designated image region and to prevent interference between conditions. In addition, we introduced a fully automated pipeline for constructing large-scale, richly annotated graphic design datasets. Extensive experiments demonstrated that CreatiDesign outperforms previous methods in subject preservation, semantic layout alignment, and overall visual quality.

**Limitation and Future Work.** CreatiDesign faces challenges in accurately preserving facial details and generating dense text, as our current dataset is not tailored for these scenarios. Improving performance in such cases, either through dataset enhancement or model-level advances, represents an important direction for future research.

## 7 ACKNOWLEDGMENTS

This work was supported by National Natural Science Foundation of China (No. 62521004).

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
