# OpenReview forum: "CreatiDesign: A Unified Multi-Conditional Diffusion Transformer for Creative Graphic Design"
_ICLR.cc/2026/Conference — ICLR 2026 Poster_

### Official Review · Reviewer_SnTD · 2025-10-30

**Soundness:** 2
**Presentation:** 3
**Contribution:** 3
**Rating:** 6
**Confidence:** 4

**Summary:**

This paper focuses on the graphic design generation task. A unified multi-condition driven architecture is proposed by integrating different user-provided elements with a multimodal attention mask mechanism. A new graphic design dataset with 400k samples featuring multi-condition annotations, along with a comprehensive benchmark. The experimental results demonstrate the effectiveness of the proposed method in adhering to user intent.

**Strengths:**

The proposed method that integrates multiple heterogeneous conditions in a unified manner for graphic design generation is reasonable and effective. The proposed large-scale, multi-element graphic design dataset could also be useful for future research in the community if released.

**Weaknesses:**

1. The current evaluations mainly compare the proposed method with existing MLLMs for controllable image generation. However, there is a domain gap between natural images and graphic designs. To make fair comparisons, existing multi-condition or single-condition graphic design generation works, e.g., [1-4], should also be discussed and compared.
2. The user study may not sufficiently demonstrate the practical effectiveness of the proposed method. The motivation of this paper is to ensure the strict adherence between the generated design and the user intent. Thus, the user study should investigate from the above criteria and ask participants to create inputs by themselves, rather than predefining inputs and following similar evaluation metrics used in quantitative comparisons.
3. Since the proposed dataset is also the contribution of this paper, the ablated versions of the automated pipeline for graphic design dataset construction should also be studied. For example, using different methods or changing the module order for the data construction process.

[1] Hsu, HsiaoYuan, and Yuxin Peng. "PosterO: Structuring Layout Trees to Enable Language Models in Generalized Content-Aware Layout Generation." CVPR 2025.

[2] Horita, Daichi, et al. "Retrieval-augmented layout transformer for content-aware layout generation." CVPR 2024.

[3] Seol, Jaejung, Seojun Kim, and Jaejun Yoo. "Posterllama: Bridging design ability of language model to content-aware layout generation." ECCV 2024.

[4] Yang, Tao, et al. "Posterllava: Constructing a unified multi-modal layout generator with llm." arXiv preprint arXiv:2406.02884.

**Questions:**

1. For entity annotation, take Fig.2 as an example, how to separate different entities as primary or secondary visual elements? Will the generated graphic design be affected by considering different entities as primary or secondary visual elements?
2. The proposed dataset is constructed via a fully automatic pipeline. Several existing models are used for design theme generation, text layer rendering, and foreground-based image generation. How to ensure the quality of the constructed dataset without involving human annotations or double-checking?

---

> ### Author Response · Authors · 2025-11-24
> **Response to Reviewer SnTD (part1)**
>
> We sincerely thank the reviewer for recognizing the effectiveness of our unified multi-condition architecture and the value of our large-scale dataset. We address your major concerns below.
>
> >Response to Q1: Comparison with Layout Generation Works ([1-4])
>
> We appreciate the references. However, we wish to clarify that **the cited works [1-4] focus on Layout Generation (outputting coordinates), while our task is Pixel-Level Image Generation.** Due to this modality mismatch, direct quantitative comparison is impossible as layout models do not produce pixel data for image quality evaluation. Consequently, **our evaluation focuses on 12 baselines within the pixel domain to ensure a fair comparison of generative capabilities (Table 1).** We will cite these papers to discuss the difference between them in the revised version.
>
>
>
> >Response to Q2: User Study Design
>
> We thank the reviewer for this suggestion and acknowledge that an interactive user study offers value. However, **we adopted the "predefined input" strategy to rigorously quantify strict adherence, as it is essential to standardize variables and establish a fixed Ground Truth for a fair comparison.**
>
> By fixing inputs (prompt, layout, text, subject), we isolate the model's instruction-following capability from human-centric variances, such as user prompting skills, subjective aesthetic preferences, and differing levels of design expertise. This methodology allows us to objectively measure the deviation between the generated output and the specific constraints, ensuring the evaluation focuses solely on model ability.
>
> >Response to Q3: Dataset Pipeline Ablation
>
> We thank the reviewer for the suggestion. We supplement ablation studies to validate the necessity of our specific pipeline design.
>
> | Model | CLIP-I | DINOv2 | M-DINO | Spatial (Sec) | Color | Textual | Shape | Spatial (Text) | Sen. Acc | NED | IR | PickScore | Avg. |
> | :--- | :---: | :---: | :---: | :---: | :---: | :---: | :---: | :---: | :---: | :---: | :---: | :---: | :---: |
> | **CreatiDesign** | **89.39** | **86.48** | **65.75** | **78.94** | **66.02** | **66.94** | **65.82** | **56.90** | **78.30** | **94.68** | **60.02** | **21.49** | **69.28** |
> | w/o HLG and FG Gen. | 88.27 | 85.30 | 65.01 | 77.39 | 65.86 | 65.51 | 63.75 | 51.86 | 58.11 | 82.89 | 58.90 | 21.22 | 65.34 |
> | w/o VLM filter | 87.04 | 83.97 | 63.77 | 75.49 | 63.28 | 60.52 | 62.00 | 52.22 | 75.27 | 92.04 | 57.97 | 21.24 | 66.23 |
>
>  We replace our proposed "Text Layer Rendering(HLG) + Foreground-based Image Generation(FG Gen)" module with direct image generation using the base model Flux.1 dev. As shown in the table above, this approach leads to a degradation in text-related performance. This confirms that our compositional rendering approach is crucial for ensuring high-quality text rendering, which general T2I models cannot yet guarantee.
> In the third row, we remove the Vision Language Model (VLM) used to filter out low-quality segmentation proposals from GroundingSAM (e.g., fragmented masks). Without this verification step, the pipeline introduces noisy data, resulting in a decline in subject preservation and layout control.

---

> ### Author Response · Authors · 2025-11-24
> **Response to Reviewer SnTD (part2)**
>
> > Response to Q4: Entity Separation (Primary vs. Secondary)
>
> During training, we employ a random partitioning strategy. Instead of using fixed rules, **we randomly assign the objects in the annotation as either primary elements (provided as image conditions) or secondary visual elements (provided as layout conditions)**. This approach ensures CreatiDesign generates high-quality images regardless of the specific combination of input conditions.
>
> > Response to Q5: Dataset Quality Control
>
> We ensure the quality of the constructed dataset through a robust compositional generation pipeline and automated VLM filtering, as validated in our ablation study (refer to Response to Q3). Specifically, **the compositional generation guarantees text quality and overall visual aesthetics, while the VLM filtering ensures the precision of annotations.** These mechanisms collectively yield a high-quality dataset with accurate multi-conditional annotations.

---

### Official Review · Reviewer_euun · 2025-11-01

**Soundness:** 2
**Presentation:** 2
**Contribution:** 2
**Rating:** 4
**Confidence:** 4

**Summary:**

This paper presents a multi-conditioned creative graphic design method, where the conditions include semantic descriptions, positions and b-boxes of objects and texts to be placed on the image. The method encodes each condition into tokens and then employ a  multimodal attention mask to prevent the attention procedure from information confusion or leaking. Experimental results demonstrates the designed posters are in high quality.

**Strengths:**

1.  The proposed multimodal attention mask is effective in control the behavior of the diffusion model as verified in the ablation study. Both layout attention mask and subject attention mask are important to the quality of the graphical design.
2. The diffusion model trained with multi-condition dataset can automatically support editing by using the image generated in the previous steps as the condition. While the STOA T-2-I models can support editing to some extent, the controllability of the proposed method should be better by associated bounding box in the graphic design.

**Weaknesses:**

The overall pipeline is built upon masked attention mechanism. However, masked attention mechanism is widely used in model design. For example, in NLP, casual attention is often used to avoid a token taking information from future tokens. From the perspective of novelty,  I hesitate to give this paper a high score.  In addition, the advantage of the proposed method are limited to the graphical design due to the specific definition of input conditions.

**Questions:**

It is tedious for a designer to determine the position of bbox of each text in a poster,  why not directly predict the position of these boxes as did in recent CVPR papers, for instance, Unsupervised domain adaption with pixel-level discriminator for image-aware layout generation in CVPR. Overall, this paper leans towards a graphic-design rendering paper more than a design paper, since the layout should be input by a user.

---

> ### Author Response · Authors · 2025-11-24
> **Response to Reviewer euun**
>
> We thank the reviewer for recognizing the effectiveness of our multimodal attention mask and the superior controllability capabilities of CreatiDesign compared to SOTA T2I models. We address the concerns below.
>
> >Response to Q1: Concerns regarding Novelty
>
> We would like to clarify that our core contribution is a Unified Multi-Condition Architecture that solves specific integration challenges, rather than an application of masked attention. Our innovations are:
>
> **1.Systematic Novelty:**
>
> We propose multimodal sequence expansion to natively integrate diverse conditions. Unlike heavy side-branch methods, our approach is highly efficient (4.1% additional params).
>
> **2.Methodological Novelty:**
>
>  Unlike NLP's 1D causal masking, our Interference-Aware Masking addresses 2D spatial topology. It is specifically designed to resolve intra- and cross-condition interference, strictly decoupling layout and subject tokens to prevent identity corruption—a level of precision standard masking cannot achieve.
>
> **3.Dataset Novelty:**
>
> Addressing the scarcity of high-quality, multi-conditioned training data, we contribute a carefully designed automated data construction pipeline. This allows for the efficient scaling of high-quality datasets, enabling the model to learn complex composition rules and aesthetic layouts that were previously difficult to capture.
>
>
>
> >Response to Q2: Concerns regarding Scope (Graphic Design)
>
> We believe the focus on graphic design is a significant strength due to the complexity and industrial demand of the task, rather than a limitation.
> Addressing Diverse User Needs (Pro vs. General): Our method is designed to serve a wide spectrum of users.
> **For Professional Designers:** The model acts as an "AI-enhanced design assistant", strictly following explicit bounding boxes and reference images to execute precise design intent.
> **For General Users:** As noted in Appendix A.1, our model supports implicit generation. If the layout condition is not provided, the model infers reasonable element placement based on the global prompt. This flexibility ensures the method is not strictly bound by manual inputs.
>
>
> >Response to Q3: Necessity of User-Provided Layouts
>
> **Our work focuses on multi-conditioned image generation rather than layout generation.**  While layout generation benefits casual users, it relies on uncontrollable generation. In contrast, **professional design workflows require deterministic control.
> Designers often possess a specific composition or design sketch. Thus, manually defining bounding boxes is not a burden but a necessary mechanism for the precise expression of design intent.** Our model is designed to faithfully execute these specific constraints, ensuring a level of adherence that automatic predictors cannot guarantee. We will cite the suggested paper and clarify this scope difference in the revision.

---

### Official Review · Reviewer_yKCt · 2025-11-02

**Soundness:** 3
**Presentation:** 2
**Contribution:** 2
**Rating:** 6
**Confidence:** 3

**Summary:**

This paper introduces CreatiDesign, a multi-condition diffusion model designed for automated graphic design generation. ​ It integrates flexible multimodal inputs, including global prompts, rendered text, foreground visuals, background elements, and their layout positions. ​ A masked attention mechanism ensures precise control, aligning layout tokens with corresponding visual elements while preventing interference.
The authors developed a new dataset with 400K synthetically generated samples and a benchmark of 1,000 annotated samples with ground truth labels. CreatiDesign is shown to outperform existing models in various design metrics.

**Strengths:**

+ a multimodal controllable generation model developed based on Flex T2I model
+ block-wise multi-modality attention achieves both controllability and prevents information leakage
+ a new benchmark dataset with curated high quality 1k set

**Weaknesses:**

- DiT framework is similar to previous work: Art: Anonymous region transformer for variable multi-layer transparent image generation. In CVPR, 2025.
- controllability is limited, only with given images and precise layout condition, text layer is not vector font. It would be useful to allow flexible layout position in output.
- the design images in the benchmark is synthetically constructed, the quality is not guaranteed.

**Questions:**

Can this method be extended to support placing given visual elements in appropriate locations in the design? This is similar to traditional layout generation/variation problem. The visual knowledge can give an edge over geometry based methods.

---

> ### Author Response · Authors · 2025-11-24
> **Response to Reviewer yKCt (part1)**
>
> We thank the reviewer for recognizing the strengths of our multimodal controllable generation, the effectiveness of the multi-modal attention mechanism, and the value of our new dataset and benchmark. We address the concerns below.
>
> > Response to Q1: Difference from ART
>
> We thank the reviewer for bringing ART to our attention. We acknowledge that both works utilize Transformer-based architectures; however, we would like to clarify the fundamental differences between our work and ART in terms of task definition and technical mechanism:
>
> **1. Different Task Definitions (Layer Generation vs. Holistic Image Generation):**
>
> ART focuses on Multi-Layer Transparent Image Generation. Its primary goal is to decompose an image into separate RGBA layers to facilitate asset creation.
> CreatiDesign, in contrast, focuses on synthesizing a cohesive pixel-level image by seamlessly integrating heterogeneous user inputs (e.g., reference images, spatial layouts, and global prompts), rather than generating decomposed transparent layers.
>
> **2. Distinct Technical Mechanisms:**
>
>  Furthermore, our proposed Multi-modal Attention Mask is fundamentally different from the Anonymous Region Layouts in ART.
> ART uses region layouts primarily to handle layer separation and transparency.
> Our Multi-modal Attention Mask is designed to enhance conditional control precision and mitigate interference between different conditions. By explicitly constraining the attention scope, our method ensures that heterogeneous conditions guide the generation process without conflicting with each other.
>
> Therefore, although both works employ DiT-like frameworks, they address distinct problems with different technical focuses. We will cite ART and include this discussion in the revised manuscript.
>
> > Response to Q2: Regarding Vector Font and Flexible Layout
>
> **1. Regarding Vector Text:**
>
> Current text-to-image diffusion models primarily operate in the pixel space to maximize visual coherence. Generating vector fonts requires different architectures (e.g., SVG generation) and represents a distinct line of research. Our work focuses on holistic image generation—ensuring the seamless integration of various conditions—rather than specialized typography generation. We believe combining our pixel-level rendering with vector-based downstream processing is a promising direction for future exploration
>
> **2. Regarding Flexible Layout:**
>
> The core motivation of our work is to achieve high-fidelity execution of user-defined layouts. In professional design workflows, designers often require strict adherence to specific spatial constraints (e.g., avoiding occlusion of key visual elements or fitting specific templates). Consequently, we prioritized precise controllability over automated layout generation in our current framework.
>
> Furthermore, we conducted additional experiments simulating scenarios where specific sub-conditions are not provided, specifically targeting the absence of Primary Visual Elements (PVE), Secondary Visual Elements (SVE), or Textual Elements (TE). As shown in the following table, the results demonstrate that CreatiDesign maintains robust performance even under partial conditions, thereby supporting flexible input configurations.
>
> | Inference Setting | IR-Score ↑ | Visual Quality (Human) ↑ |
> | :--- | :---: | :---: |
> | **Default Multi-condition** | **60.02** | **8.92** |
> | Without PVE | 59.70 | 8.85 |
> | Without SVE | 59.85 | 8.91 |
> | Without TE | 59.87 | 8.83 |

---

> ### Author Response · Authors · 2025-11-24
> **Response to Reviewer yKCt (part2)**
>
> >Response to Q3: The Quality of Synthetic Datasets
>
> We would like to clarify that our dataset is both necessary and high-quality for the following reasons:
>
> **1. Necessity:**
>
> As discussed in Section 1 (Lines 093-096), large-scale real-world graphic design datasets with fine-grained, multi-condition annotations (pixel-level masks, separate subject images, and aligned semantic descriptions) simply do not exist.
>
> **2. Quality Assurance:**
>
> We implemented a rigorous pipeline to ensure data quality:
> **Visual Quality:** The foreground-based images (Figure 4) are generated by a model fine-tuned on high-quality professional design templates, ensuring the aesthetic appeal of the source material.
> **Annotation Filtering:** After extracting segmentation masks and bounding boxes via GroundSAM, we employ a Vision-Language Model (GPT-4V) to verify the annotation. Samples with low alignment scores or low-quality masks (e.g., fragmented masks) are filtered out.
> **Validation:** The effectiveness of this data is empirically proven. In Figure 5 and Table 1, our model (trained on this synthetic data) outperforms the base model and other SOTAs in human evaluation, PickScore and ImageReward score. This confirms that the synthetic data possesses sufficient quality to empower the model robust design ability.
>
>
> > Response to Q4: Extension to Automatic Layout Generation
>
> We agree that integrating layout generation is a valuable extension, particularly for general users who may lack design expertise. Technically, a VLM or lightweight Layout Transformer could serve as a layout planner to predict bounding boxes based on visual content, which are then fed into CreatiDesign. We consider this a promising future direction to close the loop between planning and rendering, particularly for general users.

---

### Official Review · Reviewer_U8F2 · 2025-11-08

**Soundness:** 2
**Presentation:** 3
**Contribution:** 2
**Rating:** 4
**Confidence:** 4

**Summary:**

This paper presents CreatiDesign, a unified multi-conditional diffusion transformer for graphic design generation. CreatiDesign can receive inputs from different modalities: keyword, layout, prompt, image, etc. To support the task setting, the authors develop an automated pipeline for constructing a 400K graphic design dataset with multi-condition annotations. Using the dataset, they fine-tune the FLUX.1-dev model, where they modify the original attention strategy to enable multi-condition interaction. Experimental results demonstrate CreatiDesign's superior performance over existing methods in design generation.

**Strengths:**

- The paper tackles a real-world problem with clear applications in graphic design creation.
- It contributes a large-scale dataset with multi-condition annotations.
- The curated benchmark can well evaluate the design quality by including multiple metrics covering different aspects.
- The experimental results show clear improvements of CreatiDesign over baselines.

**Weaknesses:**

1. The technical contribution of CreatiDesign is limited. The used attention masking, LoRA fine-tuning are straightforward applications of existing techniques.
2. The model setting requires users to specify exact positions for the elements, making final quality heavily dependent on user design expertise. However, most users may lack such knowledge, thereby limiting its practical applicability.
3. In the experiments, it seems that the paper only evaluates scenarios where all conditions are simultaneously provided. It would be great if the authors could conduct more experiments using the same model: (1) text-to-design generation (intention-to-design without visual inputs), (2) content-aware text layout generation (placing text on given background images), etc. Evaluating diverse conditioning inputs would demonstrate practical utility across different design workflows and strengthen the effectiveness of CreatiDesign.
4. The dataset details are not included in the paper, such as the element number, prompt length, diversity, quality, etc.
5. How does CreatiDesign's inference time compare to other methods?

**Questions:**

Please see the weaknesses.

---

> ### Author Response · Authors · 2025-11-24
> **Response to Reviewer U8F2 (Part 1)**
>
> We thank the reviewer for recognizing the practical value of our work, the contribution of our large-scale dataset and benchmark, and the superior performance of CreatiDesign over existing baselines. We appreciate the constructive feedback and offer the following responses to address the concerns raised.
>
>
> > Response to Q1: Technical Novelty & Contribution
>
> We respectfully clarify that while attention masking and LoRA are foundational techniques, our work goes beyond their naive application. Instead, we focus on the targeted adaptation of these tools to solve a specific, unsolved problem: the harmonious integration of heterogeneous elements (images, layouts, and texts) in graphic design.
>
> **1. Systematic Novelty: Multimodal sequence expansion**
>
>  Unlike side-branch approaches like ControlNet or Adapters that append heavy external modules, our primary innovation lies in expanding the multimodal sequence directly within the original architecture.
>  We extend the existing text-image token sequences to include diverse condition modalities. By seamlessly integrating these extended sequences into the original MM-Attention framework, we achieve robust multi-condition control capabilities. This native integration approach is highly efficient, requiring only 4.1% additional parameters.
>
> **2. Methodological Novelty: Interference-Aware Masking**
>
>  Our masking strategy is fundamentally different from generic implementations. It is driven by our insight into two specific challenges: intra-condition interference (conflict between similar elements) and cross-condition interference (conflict between layout and subject identity).
>  We designed a tailored multi-modal attention mask that strictly decouples these interactions. This ensures that each sub-condition retains its independent characteristics and precisely controls its corresponding image region without corrupting others—a level of control that standard masking cannot achieve.
>
> **3. Dataset Novelty: Automated Pipeline for Scalability**
>
>  Addressing the scarcity of high-quality, multi-conditioned training data, we contribute a carefully designed automated data construction pipeline. This allows for the efficient scaling of high-quality datasets, enabling the model to learn complex composition rules and aesthetic layouts that were previously difficult to capture.
>
>
>
> >  Response to Q2: Concern regarding the dependency on design expertise
>
> We would like to clarify that **CreatiDesign accommodates both professional workflows and general users through a flexible input mechanism.
> For professional designers, the ability to specify exact positions is not a burden but a critical requirement. A designer typically starts with a specific composition or a rough sketch and requires a model to accurately visualize that specific intent.
> This scenario—strict adherence to complex spatial constraints—is significantly more challenging than unconstrained generation.** Previous models often fail to respect these specific layouts, deviating from the designer's intent. Our paper primarily targets this unsolved challenge, providing the high-fidelity control necessary for professional graphic design.
> For general users, CreatiDesign supports generation when some sub-conditions are not provided (see Appendix A.1), which reduces the design burden by allowing partial inputs.
>
>
> >  Response to Q3: Evaluation on Diverse Conditioning Inputs
>
> We thank the reviewer for this constructive suggestion. In Appendix A.1, we discussed that our model can generate high-quality and coherent designs even when provided with only one type of control (e.g., subject image or semantic layout).
>
> To further support this quantitatively, **we conducted additional experiments simulating scenarios where specific sub-conditions are not provided**, specifically targeting the absence of Primary Visual Elements (PVE), Secondary Visual Elements (SVE), and Textual Elements (TE). These settings correspond to the flexible workflows mentioned by the reviewer. As shown in the following table, the results demonstrate that CreatiDesign maintains stable and high performance even under partial conditions.
>
>
> | Inference Setting | IR-Score ↑ | Visual Quality (Human) ↑ |
> | :--- | :---: | :---: |
> | **Default Mutil-condition** | **60.02** | **8.92** |
> | Without PVE | 59.70 | 8.85 |
> | Without SVE | 59.85 | 8.91 |
> | Without TE | 59.87 | 8.83 |

---

> ### Author Response · Authors · 2025-11-24
> **Response to Reviewer U8F2 (Part 2)**
>
> >Response to Q4: Dataset Details
>
> Thank you for your suggestion. In the revised manuscript, we have added a comprehensive analysis of the dataset statistics, diversity, and quality assurance mechanisms.
>
> **1. Quantitative Statistics:**
>
>  We have conducted a statistical analysis on the 393,738 samples in the CreatiDesign dataset, as detailed in the newly added Figure 10 and Appendix A.3. CreatiDesign averages 3.60 annotations and 1.50 text instances per sample. Regarding prompt length, the global captions are categorized into Detailed Global Captions, which average 92.77 tokens (Median: 93.00), and Short Captions, which average 15.74 tokens.
>
> **2. Diversity:**
>
>  As illustrated in Figure 4, the Design Keywords are drawn from an open-world vocabulary, covering a vast range of concepts. Furthermore, the Design Theme Generation module produces highly diverse descriptions, preventing repetitive patterns.
>
> **3. Quality Assurance:**
>
> To guarantee high quality in both images and annotations, we utilize Text Layer Rendering combined with a pretrained Foreground-based Image Generation model to ensure that the generated designs are visually coherent and high-fidelity. Besides, we employ a VLM to filter out mismatched or low-quality data, thereby ensuring precise alignment between the images and their annotations.
>
> >Response to Q5: Inference Time
>
> Evaluated on a single NVIDIA A100 GPU, the inference time of CreatiDesign for generating one image is 18 seconds. This is comparable to baselines such as PosterMaker (13s) and UNO (17s), and faster than others like BizGen (72.5s) and HiCo (137s).

---

### Meta-Review · Area_Chair_adAD · 2025-12-22

**Summary:**

The major concerns of the reviewers are: 1) the technical novelty is limited; 2) the flexibility of the model is not high. I partially agree with the two points. However, the paper solved an important problem that hasn't been well studied previously, and the proposed strategy is promising. Therefore, I recommend accepting the paper.

**Reviewer Concerns:**

**Reviewer U8F2**'s major concerns are as follows:
1. The technical contribution of CreatiDesign is not significant.
2. The model setting requires domain knowledge and is not flexible.
3. The experimental evaluation is not comprehensive.

I think these three points have been addressed to some extent by the rebuttal.

**Reviewer yKCt**'s major concerns are as follows:
1. The novelty is limited, especially when compared to a paper in CVPR 2025.
2. The controllability is limited.
3. The design images in the benchmark are synthetically constructed.

These concerns have been addressed to some extent by the rebuttal.

**Reviewer euun**'s major concern is about the novelty of the proposed model, which hasn't been addressed.

**Reviewer SnTD**'s major concerns are as follows:
1. The comparison is not fair. Some multi-condition or single-condition graphic design generation works should also be discussed and compared.
2. The user study may not sufficiently demonstrate the practical effectiveness of the proposed method.
3. The ablation study is insufficient.

I think that concerns 2 and 3 have been addressed by the rebuttal.

**Reviewer Scores:**

If there was a full discussion, I think that Reviewers yKCt and SnTD will keep their positive ratings (6), Reviewer euun may keep the negative rating 4, and Reviewer U8F2 may increase the rating from 4 to 6.

---

### Decision · Program_Chairs · 2026-01-26

Accept (Poster)